# Thermal Processing of Liquid Egg Yolks Modulates Physio-Chemical Properties of Mayonnaise

**DOI:** 10.3390/foods11101426

**Published:** 2022-05-14

**Authors:** Jou-Hsuan Ho, Tan-Ang Lee, Nobuaki Namai, Shunji Sakai, Siao-Syuan Lou, Akihiro Handa, Wan-Teng Lin

**Affiliations:** 1Department of Food Science, Tunghai University, Taichung 40704, Taiwan; jhho@thu.edu.tw (J.-H.H.); danyanlee@gmail.com (T.-A.L.); cc5cc57788@gmail.com (S.-S.L.); 2R&D Division, Kewpie Corporation, Tokyo 182-0002, Japan; nobuaki_namai@kewpie.co.jp (N.N.); shunji_sakai@kewpie.co.jp (S.S.); 3Division of Life Science and Engineering, School of Science and Engineering, Tokyo Denki University, Saitama 350-0394, Japan; ahanda@mail.dendai.ac.jp; 4Department of Hospitality Management, College of Agriculture, Tunghai University, Taichung 40704, Taiwan

**Keywords:** liquid egg yolk, mayonnaise, heat treatment, protein denaturation, viscosity, emulsification stability, particle size

## Abstract

In this study, the effect of various heating temperatures (61–70 °C) and times (1–10 min) on physical and chemical properties of liquid egg yolk (LEY) and mayonnaise were investigated. Initially, we found that the increase of LEY protein denaturation was highly correlated with the increase of temperature and time, without causing either protein degradation or aggregation. In addition, the viscosity and particle size of LEY were significantly increased with greater heating temperature and time. Furthermore, the emulsification stability of mayonnaise prepared from thermally processed LEY were significantly better than that of the unheated control group, in particular, the emulsion stability of mayonnaise was higher at a temperature ranging from 62 °C to 68 °C, whereas the emulsion stability decreased above 69 °C. A rheological analysis showed that mayonnaise prepared from thermally processed LEY has higher shear stress when compared with the control group. Indeed, a sharp increase in the shear stress was observed when LEY was heated above 67 °C. Results from storage behavior analysis suggest that mayonnaise prepared from thermally processed LEY failed to affect the chemical qualities of mayonnaise, as evidenced by the fact that acid values and TBA values were not statistically significant with the unheated control group. Microscopic observation indicates that the number of complete oil droplets were significantly reduced at higher heating (70 °C/5 and 10 min) conditions. Finally, the sensory evaluation results suggest that mayonnaise prepared from thermally processed LEY does not influence the appearance, aroma, taste, greasy feeling, and overall acceptance of mayonnaise, as indicated by there being no significant differences between the experimental group and the control group (*p* > 0.05). We conclude from our study that a combination of heating conditions over 67 °C/5 min can allow the mayonnaise to retain better quality in terms of stability.

## 1. Introduction

Eggs are regarded as one of the best sources of high-quality protein, and they are considered to be a nutritious food. Besides being a natural food emulsifier, egg yolk is also the primary ingredient in mayonnaise and a variety of other salad dressings and sauces [1]. As egg yolk is extremely sensitive to microbiological contamination, it is essential to thermally treat it in order to ensure its innocuous composition [2]. The process of pasteurizing egg yolks for industrial use typically involves heating at temperatures between 60 and 68 °C from a couple of seconds to about 10 min, depending on the temperature. The purpose of this treatment is to eliminate pathogenic microorganisms such as *Salmonella* without causing damage to the egg yolk proteins or their functional properties [3]. Nevertheless, it is not able to completely eliminate the microbial flora from egg yolk, thus it only has a limited shelf life and must be stored at 4 °C. In order to further ensure egg microbial safety and to increase shelf life, some egg processors and users would prefer to use stronger heat treatments. However, some of the components in egg yolk are thermally sensitive, therefore such a heat treatment may alter their physical and functional properties [4].

Structurally, egg yolk is a solution comprising soluble glycoproteins called livetins and low-density lipoproteins (LDL). A typical LDL consists of around 12% proteins and 87% lipids and presents a spherical shape with an average diameter of approximately 35 nm [5]. The monolayer consists of phospholipids encapsulated by apolipoproteins and is composed primarily of triglycerides, cholesterol, and cholesteryl esters. Apolipoprotein (ApoLDL), the main component of LDL that is involved in adsorption at the oil-water interface, forms thin films around oil droplets by adsorbing at the oil-water interface [5]. It has been demonstrated that these components denature at temperatures as low as 62 °C to 65 °C [6]. Gelation of LDL solutions begins to occur at low temperatures (starting at 69 °C); denatured LDL solutions begin to gel at 70 °C, and solid LDL solutions gel at 75 °C [7]. A previous study has shown that LDL may play a significant role in egg yolk emulsion formation and stabilization [8]. Apolipoproteins from egg yolks form a robust membrane to surround lipid droplets, in which triglycerides from oil interact hydrophobically with amino acids from apolipoproteins in egg yolks. Kiosseoglou and Sherman [9] proposed that these triglycerides hydrophobically bind to apolipoproteins in oil droplets. LDL micelles are broken when they contact the oil-water interface, the lipid cores combine, the apoLDL and phospholipids diffuse, and a film is formed that stabilizes the emulsion by protein adsorption on the interface.

The mayonnaise is a product that illustrates the oil-in-water (O/W) emulsion process and has a highly concentrated internal phase. Although several food emulsions are thermodynamically unstable, they tend to coalesce, flocculate, and cream, while mayonnaise has a relatively long shelf-life of over a year with no change in its organoleptic properties [10]. This is believed to be due to the fact that it has a high viscosity resulting from the close packing of droplets. However, the precise mechanism by which mayonnaise is destabilized remains unclear. 

In many studies, mayonnaise has been used as a model system to study the emulsifying properties of egg yolks. Mayonnaise is an oil-in-water emulsion with a high oil content that manifests its unique rheological properties and texture as a result of its microstructure [11]. The general consensus is that mayonnaise exhibits pseudoplastic properties that include a flow threshold and thixotropy. There has been a lot of attention paid to the rheological behavior of mayonnaise, which has been studied using steady shear rotational rheology, small amplitude oscillatory shear, or even both techniques. Additionally, the perceived texture and the emulsifying mechanism of egg yolk in mayonnaise has been investigated. Few studies are available to indicate that egg yolk pasteurization does not affect the stability of O/W emulsions, although little attention has been paid to understanding the effect of thermal treatment on egg yolk’s emulsifying properties [12,13]. A previous study demonstrated that thermal treatment of about 3 min at temperatures as high as 76 °C led to a slight decrease in emulsifying activity but no modification in the emulsion stability of egg yolk [14]. The results of another study by Campbell et al. [12] showed that heating egg yolk at 77 °C for 2 min had no impact on its emulsifying activity in concentrated emulsions, while they did not describe the rheological properties of the emulsions. In addition, Guilmineau and Kulozik [15] reported that the heating of hen’s egg yolk prior to emulsification significantly impacts the rheological properties of mayonnaise when the median oil droplet diameter exceeds 5 μm. However, there are no studies that examine the physical and chemical properties of mayonnaise prepared from thermally processed LEY. Therefore, the present study was aimed to investigate the emulsion, rheological, storage and sensory properties of mayonnaise prepared from thermally treated LEY and to optimize suitable sterilizing temperature and time to the quality of mayonnaise.

## 2. Materials and Methods

### 2.1. Liquid Egg Yolk (LEY) Sample Preparation

A fresh batch of eggs from hens of the leghorn were collected from a standard supermarket (PX Mart, Taichung, Taiwan). Egg yolks were separated manually by tapping on the eggshell. Egg whites and yolks were separated manually. In order to ensure that there was no egg white remaining in the egg yolk, yolks were gently rolled on the filter paper to remove residues of egg whites and then the yolk membrane was carefully broken to collect the yolk liquid. Twenty grams of LEY were poured into a high-density polyethylene bag and tightly sealed. A self-made constant temperature circulating water tank was used to attain the target temperature and time at a heating rate of 11 s/°C. In order to maintain uniformity, the initial temperature of the constant temperature circulating water tank is 25 °C, and the water volume is 4 L. The LEY samples were treated with various experimental temperature (61–70 °C) and time (0, 1, 4, 5, 7, 10 min), samples were immediately cooled down with ice cold water, and then stored in a refrigerator for further experiments.

### 2.2. Physicochemical Analysis of LEY

#### 2.2.1. Degree of Protein Denaturation

The degree of protein denaturation was determined by adopting a previously published protocol [16]. Briefly, heat treated LEY samples were dissolved in 0.1 M phosphate buffer (pH 7.5 ± 0.1) to obtain a total protein concentration of 800 μg/mL. The samples were then left to equilibrate at a temperature of 20 °C for 1 h with gentle agitation. The sample was divided into 2 aliquots with equal volume, one for the measurement of total protein content (Pt), and the other was centrifuged twice at 19,000× *g* for 20 min. Soluble protein content (Ps) in the supernatant was determined by a bicinchoninic acid-protein (BCA) assay kit (Abcam, Cambridge, UK). Denaturation of LEY proteins were calculated as follows:(1)$$\mathrm{Dp}\%=100-(\frac{\mathrm{Ps}}{\mathrm{Pt}}\;\times \;100\%)$$

#### 2.2.2. Particle Size Measurement of Liquid Egg Yolk

Fresh as well as heated egg yolks were measured for particle size distribution using a particle size analyzer (LA-960, HORIBA, Kyoto, Japan). A homogeneous sample was prepared by diluting the samples with deionized water at a ratio of 1:1.5 (*v*/*v*) on a stir plate for 1.5 h. A triplicate measurement was performed when the shading ratio reached 5–10%. There were two refractive indices of 1.33 for the water/background and 1.46 for the yolk/sample. Particle diameters between 0% and 50% were added to create the mean particle size (D50).

#### 2.2.3. Determination of Viscosity of Liquid Egg Yolk

Viscosity was determined using a rheometer (R/S+ Rheometer, Brookfield, Temecula, CA, USA). In order to determine the viscosity of the samples at 25 °C, a parallel stainless-steel plate with a diameter of 1.5 mm and a gap (Gap) of 0.45 mm was used. Initially, the shear rate was set at 0 to 10 s^−1^ (or 100 s^−1^) for 2 min, then held at 10 s^−1^ (or 100 s^−1^) for 2 min, and finally reduced to 0 s^−1^ for 2 min. The viscosity of LEY is calculated from the average of 12 data points per point for every 10 s while maintaining a shear rate of 10 s^−1^ (or 100 s^−1^)/2 min.

Thixotropic loops were performed by first increasing the shear rate logarithmically from 0 to 100 s^−1^ over 2 min, then holding it at 100 s^−1^ for 2 min, and finally reducing it to 0 s^−1^ over 2 min. 

We determined the shear stress, viscosity, flow behavior index, and thixotropy using the Herschel-Bulkley equation model as follows:τ = τy + K × γ^n^(2)
where τ is the shear stress (Pa), τy is the yield stress (Pa), γ^n^ is the shear rate (s^−1^), K is the consistency index (Pa s^−1^), and n is the flow index.

#### 2.2.4. LEY Protein Sample Preparation and SDS-PAGE

Pretreatment and LEY protein samples were prepared according to previously published method [16]. Briefly, LEY samples were dissolved in 50 mM Tris–acetate buffer (buffer containing 1% *w*/*v* SDS, 8 M urea and 50 μg/mL bromophenol blue), and then samples were continuously shacked for 12 h to completely dissolve the solution. Samples were heated at 100 °C for 5 min, and centrifuged at 10,000× *g* for 10 min at room temperature. An equal amount of protein samples (30 μL) underwent sodium dodecyl-sulfate polyacrylamide gel electrophoresis (SDS-PAGE). Protein samples were electrophoresed with an initial power supply at 90 v for 15 min and then at 120 v for 80 min. After the protein samples were separated, SD-PAGE was stained with Coomassie Brilliant Blue (CBR) for 40 min. The gels were fading from the alcohol, and pictures were immediately taken.

#### 2.2.5. Response Surface Methodology (RSM)

Egg yolks were heated at temperatures ranging from 61 °C to 70 °C for 1 min to 10 min according to the range established by the experimental design. Data were analyzed by STAGRAPHICS statistical software (Stagraphics Technology, Inc., The Plains, VA, USA) to obtain a set of two equations, and a response surface graph was made.

### 2.3. Physicochemical Analysis of Mayonnaise

#### 2.3.1. Mayonnaise Sample Preparation

The mayonnaise was prepared according to a previously published protocol [17] with minor modifications. A HD-002 homogenizer along with No-13 blade (Shin Kwang Machinery Industry Co., Ltd., Taipei, Taiwan) was used to prepare mayonnaise. A heated liquid egg yolk (14.05%) was placed in a beaker, edible vinegar (9.20%) was added, and the mixture was homogenized for 10 s at 1000× *g*. Then, white sugar (2.70%) and salt (1.05%) were added to the sample and homogenized for 1 min at 2000× *g*. Finally, soybean oil (73%) was added at a constant rate within 1 min and homogenized at 3000× *g* for 2 min to complete the process.

#### 2.3.2. Determination of Mayonnaise Emulsion Stability

The stability of mayonnaise was determined by a previously published protocol [18]. Briefly, a 5 g mayonnaise sample (F0) was placed in a centrifuge tube, sealed with a plastic cap and stored at 50 °C for 48 h, and then centrifuged at 3000× *g* for 10 min, and oil from the surface was removed. After centrifugation and removal of the slick, the weight of the mayonnaise (F1) was determined. The emulsion stability of mayonnaise was calculated using the following formula.
Emulsion stability (%) = (F1/F0) × 100(3)

F0: sample weight; F1: sample weight after removing the oil slick.

#### 2.3.3. Rheological Tests of Mayonnaise

Rheological measurements were performed with a rheometer (R/S+ rheometer, Brookfield Engineering Laboratories, Inc., Middleboro, MA, USA). A parallel stainless-steel plate with a diameter of 49.95 mm was used to examine the flow properties of mayonnaise at 25 °C. The thixotropic loops were performed by increasing the shear rate logarithmically from 0/s to 100/s over 2 min, then maintaining it at 100/s for an additional 2 min, and finally reducing it to 0/s over 2 min. Shear stress, viscosity, flow behavior index and thixotropic data were obtained by the Herschel-Bulkley equation model as follows:τ = τy + K × γ^n^(4)
where τ is the shear stress (Pa), τy is the yield stress (Pa), γ^n^ is the shear rate (/s), K is the consistency index (Pa·s^−1^), and n is the flow index.

#### 2.3.4. Particle Size Measurement of Mayonnaise

The particle size distributions of the prepared mayonnaise were determined by applying a particle size analyzer (LA-960, Horiba Ltd., Kyoto, Japan). Sample mayonnaise (0.04 g) was diluted to an appropriate concentration with 150 mL of 0.1% sodium dodecyl sulfate (SDS) solution and stirred gently with a spatula in order to fully disperse the mayonnaise. The sample was dispersed in distilled water until it created 5–10% shadowing, and the background and sample were measured. The particle size distribution was then calculated using Mie theory at a refractive index 1.460. The particulate diameter from 0% to 50% was used to determine the mean particle size (D50). 

#### 2.3.5. Scanning Electron Microscope (SEM) 

A previously published protocol [19] was used to prepare mayonnaise samples for SEM. Briefly, mayonnaise samples were encapsulated in 2.5% agar and fixed overnight in 2% glutaraldehyde in a 0.1 M phosphate buffer (pH 7). Mayonnaise samples were dehydrated with a graded series of alcohol. Dehydrated samples were critical point dried through CO_2_ in a critical point dryer (SamdiPVT3, Tousimis, Rockville, MD, USA). Microcapsule samples were placed on aluminum sample stages and covered with double-sided tape. The aluminum stage was then placed into an ion laminator (JFC-1600, JEOL, Tokyo, Japan) for gold plating. Gold-plated samples were placed in the sample chamber of the SEM (JEOL), and the microstructure of the mayonnaise was observed at 2000× magnification under a vacuum and photographed for structure comparison.

#### 2.3.6. Storage Analysis of Mayonnaise

##### Preparation of Oil Fraction from the Mayonnaise

A 25 g mayonnaise sample was taken and placed into a centrifuge tube and kept in a −20 °C refrigerator to freeze overnight. The next day, the sample was taken out and thawed at room temperature for 30–60 min, then centrifuged at 3000× *g* for 10 min at 28 °C. The upper oil layer was collected and ready for further analysis.

##### Determination of Acid Values (AV)

According to the method of The American Oil Chemists’ Society (AOCS, Cd38-63), a 5 g oil sample was put in a conical flask and 15 mL of alcohol-ether mixture (1:1, *v*/*v*) was added, and the samples were completely dissolved. On the other hand, a blank test sample (alcohol-ether mixture without oil sample) was prepared. Two to three drops of phenol indicator was added to the sample and blank solutions, and then 0.05 N potassium hydroxide solution was added for titration; the end point of the titration was considered to be when the test solutions turned pink and the color remained for 30 s.
AV (mg KOH/g) = (V1 − V2) × N × 56.11/W(5)

V1: The volume of potassium hydroxide titrant consumed by the sample solution (mL)

V2: The volume of potassium hydroxide titrant consumed by the blank test solution (mL)

N: concentration of potassium hydroxide 

W: sample weight (g).

##### Determination of Thiobarbituric Acid (TBA) Values 

Peroxidation of lipid molecules were determined by the TBA method as described previously [20]. Briefly, 3 g of oil samples were placed into a test tube, and 10 mL of benzene solution and 12 mL of TBA reagent was added. Test tubes were tightly sealed and shaken for 2 min, and then left to stand for stratification. After the layers were separated, the water layer was collocated and transferred to a clean test tube and it was heated in a boiling water bath for 30 min. Next, the samples were cooled and the peroxidation was measured with an absorbance at 530 nm.
TBA value (μg/g) = [(A1 − A2)/155/W] × 1000(6)

A1: Absorbance value of the sample

A2: Absorbance value of blank test solution

W: sample weight

Number of absorbance bars: 155 mM-’/cm.

### 2.4. Sensory Evaluation

Sensory evaluation was carried out through quantitative descriptive analysis (QDA) by a panel consisting of 60 assessors. The panelists were students aged between 18 and 25 from the Department of Food Science, Thunghai University, Taiwan. Brief training was conducted before the analysis in order to verify and explain the vocabulary, as well as the scales used. The samples are numbered randomly with three random numbers, and they are randomly arranged in order to reduce sample arrangement errors. The samples were given to eat with freshly baked white toast at room temperature, plain soda crackers, and drinking water was provided for the tester to deodorize them between sample tasting. The following attributes were evaluated: appearance, aroma, mouth feel, greasy feel and overall acceptability. The ranking was defined as follows: 1 = lowest intensity and 9 = highest intensity. Data were normalized to the average score given by each panelist for each attribute. Then, data were recalculated using the calibration curve obtained by plotting each attribute’s raw data (*y*-axis) versus normalized data (*x*-axis). Through this approach, differences in the scale of values of each non-calibrated panel member were avoided. The average scores for each attribute were determined and reported.

### 2.5. Statistical Analysis

The experimental results were expressed as mean ± standard deviation (mean ± SD) from two to three independent experiments. Statistical analysis was performed using one-way ANOVA followed by Dunnett’s test for multiple comparison. A statistical significance of *p* < 0.05 was used as the criterion. 

## 3. Results and Discussion

### 3.1. Thermal Processing Influences the Degree of LEY Protein Denaturation

The effect of temperature and time on the denaturation of LEY protein is shown in Table 1. A total of 61 condition groups were performed, which ranged from 61 °C to 70 °C, and a total of three replicates were performed for each condition. We found that the degree of denaturation of LEY protein increases with ascending heating temperature and time, and in all experimental conditions the degree of denaturation is greater than that of the control group. The combined results of 61~70 °C and 0, 1, 4, 5, 7, and 10 min are shown in Table 1. According to the results, the standard deviation of the results is wider at the lower heating condition (61 °C) compared to the higher temperatures. In addition, it can be observed that the degree of denaturation of LEY after being heated at 65 °C, especially the heating time of 10 min, has increased significantly. It was observed that a significant increase in protein denaturation was observed under the heating condition of 68 °C/5 min compared to the 68 °C/4 min condition. Indeed, at higher temperatures, shorter heating time produced the same degree of protein denaturation, e.g., 70 °C/1 min is similar to 61 °C/10 min. In egg yolks, LDL apolipoprotein and vitelloglobulins are two of the most heat-sensitive proteins [7]. It has been demonstrated that these components frequently denatured at temperatures as low as 62 °C to 65 °C [21]. Observations in this study found that, regardless of temperature and time, the degree of denaturization of egg yolks increased. Additionally, the difference may influence the degree of denaturation of LEY proteins. The corresponding western blot data are shown in Appendix A.

### 3.2. Thermal Processing Increased LEY Protein Particle Size

A protein’s particle size is one of its most important physicochemical properties, because the particle size can directly influence its interfacial properties such as foaming, emulsification, and other functions [22]. Table 2 indicates the particle size distribution of egg yolks heated at 61~70 °C for different time points (1, 4, 5, 7, and 10 min). The unheated fresh egg yolk was served as a control group, and the mean particle size (D_50_) of the control group was recorded as 6.12 μm. The mean particle size of LEY is well correlated with the other observation [23] that the untreated fresh egg yolk may have particle size ranging from 2.5 to 50 μm. It was well demonstrated that egg yolk particle size increased according to heating temperature and time under experimental conditions [24]. In egg yolks, there are a variety of proteins that have different chemical structures and thus varying levels of heat sensitivity. At any given temperature, the constituent proteins denature at varying rates. The denaturation of proteins during heating leads to the formation of insoluble aggregates [15]. Even with the same heating temperature, the particle size distribution after heating for 1 min will be different from that after heating for 10 min. Based on our data, we found that the average particle size (D_50_) obtained by heating for 10 min was more than that of heating for one min in 61 °C, 66 °C), 67 °C, and 70 °C, which indicate that the average particle size increased with the increase of heating time (Table 2). Nevertheless, it is noteworthy that under certain heating conditions, particle size is smaller than the average particle size. Notably, the particle size appears considerably smaller in the 62 °C, 63 °C, 64 °C, 65 °C, 68 °C, and 69 °C (Table 2) treatment groups, respectively. Under the heating time of 1 min and 10 min at 64 °C, the average particle size (D_50_) is 15.47 μm and 23.2 μm, respectively, which is much smaller than the average particle size under the heating times of 4 min and 7 min at 108.47 μm and 95.99 μm, respectively.

In egg yolks, the gel structure develops gradually, and its structural consolidation is the result of the gradual participation of the network of yolk protein components, which have different molecular structures under varying heating conditions [25]. It can also be explained that the egg yolk will behave differently after different heating conditions and temperatures. According to Kiosseoglou and Paraskevopoulou [26], the binding forces of proteins in heat-treated egg yolks are predominantly hydrophobic interactions, which are weak and can be disrupted by shear forces under emulsification conditions. Another study by Suhag et al. [27] reported that the particle size test of egg yolk was conducted without any sonication and stirring to prevent protein denaturation, which may suggest that egg yolk particles will be smaller under certain conditions.

### 3.3. Heating Time and Temperature Increased LEY Viscosity

The viscosity of egg yolk under steady shear conditions with a shear rate of 10/s and 100/s at 61~70 °C and 1, 4, 5, 7, and 10 min of heating conditions were determined. As the temperature is raised to 65 °C, the viscosity begins to increase relatively slowly within 4 min. At a heating time between 4 and 10 min, a temperature of 66 °C is shown to greatly increase the viscosity, especially that occurring between 68 °C and 70 °C. The egg yolk viscosity at 66~70 °C has a higher numerical value under the heating time conditions of 7 min and 10 min. Figure 1a clearly illustrates the combination diagram of egg yolk at 61~70 °C, for 1, 4, 5, 7, and 10 min under constant shear rates of 10/s, which reveals the difference between each temperature viscosity under a variety of heating time conditions. In terms of a heating time of 1 min, except at 70 °C, the viscosity values are almost similar and did not exceed 5 Pa·S; also there is no noticeable difference. Figure 1b shows the viscosity values of egg yolk at 61~70 °C under constant shear rates of 100/s under heating conditions of 1, 4, 5, 7, and 10 min. When compared to the shear rate of 10/s, the gap between 100/s and 10/s is smaller, but after heating at 66 °C, there are obvious differences between 1 and 4 min and 5, 7 and 10 min. In addition, we observed that egg yolks thicken around 65 °C during gradual heating, whereas they become solid at 70 °C, a reaction may be caused by the hypodenaturation of globulin, which is present in the egg yolk [26]. Furthermore, the gelation of yolks reflects a reduction in the stability of LDL due to the unfolding of its apolipoproteins, resulting in molecular interactions and ultimately in the formation of inter-particle gels [28]. It was observed that the initial phase of heating at high temperatures (68 °C for 1–4 min), corresponded to the time required to expose egg yolk apolipoproteins to heat, which was characterized by certain changes. In response to egg yolk denaturation reaching a critical concentration, inter-particle gels began to interact with one another, resulting in a substantial increase of viscosity.

### 3.4. Response Surface Pattern Analysis

The response surface pattern can be derived from the statistical regression of the experimental data using a statistical software package which can be used to determine the correlation of the data. An analysis of variance (ANOVA), coefficient of determination (R^2^) and ratio of variance test (F-test value) were used and the reaction surface graph was generated. R^2^ values closer to 1 indicate that the experimental data are significant and of a lower standard deviation, while the R^2^ value is much lower than 1 indicated a higher standard deviation in between each experimental group. Response surface pattern analysis revealed that LEY heating times and temperatures were highly correlated with degree of protein denaturation (R^2^ = 0.88) (Appendix A), while there is a moderate correlation between LEY heating times and temperature for viscosity (R^2^ = 0.61) (Appendix A), and barely correlated with particle size (R^2^ = 0.09) (Appendix A). We mentioned in our previous particle size results that protein-binding interactions are very weak and can be disrupted by other forces, thereby causing particle size to change, which may also explain the reduced correlation.

### 3.5. Effect of Thermally Processed LEY on Emulsification Stability of Mayonnaise

The effect of emulsification stability of mayonnaise prepared from thermally processed LEY were examined. In nature, yolk granules are composed of insoluble compounds such as high-density lipoprotein (HDL) and phosphorescent proteins that can form a stable oil-in-water (O/W) emulsion when mixed with water [8]. Shariful et al. [29] demonstrated that the size distribution of the oil droplets in the mayonnaise changed over the course of time, which resulted from the phenomenon that the oil droplets in the mayonnaise became smaller or larger, thus separating the water phase and oil phase of the mayonnaise. In contrast, emulsion stability involves preventing droplet coalescence, flocculation, and emulsion. As a result of this study, we observed that mayonnaise prepared with egg yolks thermally treated at 64 °C for 1 min or more had a relatively high emulsion stability (Figure 2). Indeed, mayonnaise made from egg yolks thermally treated with 70 °C has relatively stable emulsification stability (Figure 2). The emulation stability for all other experimental groups was shown in Appendix A. This observation is well correlated with Guilmineau and Kulozik [15], who reported that emulsions made with untreated egg yolks tend to flocculate more than emulsions made with heated egg yolks, and heat-denatured egg yolks demonstrate comparable emulsifying activity. Our data suggest that under the heating condition of 64 °C for 1 min or more, the emulsification stability was increased by more than 20% than that of the control group. In addition, higher heating temperature and longer heating time allow the emulsification stability of mayonnaise.

### 3.6. Effect of Thermally Processed LEY on Mayonnaise Steady Sate of Rheology

Mayonnaise samples prepared from thermally processed LEY showed a non-Newtonian pseudoplastic behavior. As shown in Figure 3, the shear rate of mayonnaise increases with the increase of temperature at which LEY is heated. Control mayonnaise samples showed a lower shear stress with increasing shear rate (Figure 3a). At the lower heating temperature of 64 °C, it was noted that there was no significant difference in the shear rate among the experimental groups at the same temperature and at different time points. In contrast, the mayonnaise under the heating condition of 64 °C/10 min showed a large difference from that of 64 °C/1 min and 64 °C/5 min (Figure 3b). At the higher temperature of 70 °C, large and obvious differences are evident under different time conditions. The temperatures of 70 °C/1 min, 70 °C/5 min, and 70 °C/10 min are concentrated under extremely high shear stress (Figure 3c). The shear stress and shear rate for all the tested temperatures and time are shown in Appendix A. Changing rheological properties at high temperatures may be caused by an increase in thermal energy, leading to the breakage of the interfacial film at the droplet interface.

### 3.7. Effect of Thermally Processed LEY on Mayonnaise Particle Size

Mayonnaise is a typical O/W emulsion, and the particle size of the emulsion plays a functional role in its stability and performance in food applications [30]. Studies have shown that in condiments such as mayonnaise or salad dressing, smaller particle sizes of emulsions produce a more delicate taste and a better mouthfeel [31]. Therefore, we examined whether the thermal processing of LEY may affect the mayonnaise particle size. Based on our result, we found that the particle size of all the experimental groups were below 4 μm, whereas the D_50_ was greater than 5 μm in 70 °C/1 min treatment group. It is evident that the mayonnaise made from egg yolks heated at a higher temperature has smaller oil droplets (Table 3). In addition, it was found that the average mayonnaise particle size of all experimental groups is below 7 μm and around 5 μm, which is similar to the commercially available mayonnaise products [32]. Theoretically, it was demonstrated that a reduction in the diameter of the oil droplets results in a larger contact area between the oil droplets, and therefore provides a greater viscosity [33]. There is no correlation between the particle size of this experiment and the rheological test; however, the results are highly correlated with other observations [34]. For example, Liu and co-workers [34] reported that the experimental group with the largest particle size had lower values for hardness, consistency, cohesion, and viscosity than the control group.

### 3.8. Microstructural Characterization of Mayonnaise

Mayonnaise is largely composed of organic carbons; thus, it has weak electrical conductivity, which requires a multi-step pretreatment process to ensure smooth electrical conductivity. Furthermore, mayonnaise is an oil-in-water emulsion. An emulsion of this type consists of an emulsifier, a continuous liquid phase containing water and water-soluble components, and oil and oil-based components as a dispersed phase. Under a scanning electron microscope, there are round spheres that can be observed surrounding a nearly transparent film, which signifies that the water phase of the mayonnaise is surrounded by oil droplets (Figure 4). It is noticeable that oil droplets are very uniform in size, all are less than 10 μm in either the control or treatment groups. Notably, oil droplets of different sizes can be observed at 64 °C/10 min, and some oil droplets appear irregular (Figure 4e). There is a possibility that, in mayonnaise, the high oil content and the close packing of the oil droplets may affect the oil droplets, causing them to deform from their normal sphere shape [35]. Due to the irradiation of electron beams, it becomes blurry during the 70 °C/10 min imaging period. The electron microstructure image looks more like oil droplets embedded in a body. Despite a larger oil droplet in the center, all of the other circular protrusions are smaller than the control. According to the electron microstructure, it can be expected that the size of the complete oil droplets that can be separated at 70 °C/10 min will be larger (Figure 4i), along with the particle size increasing when the particle size is detected.

### 3.9. Chemical Properties of Mayonnaise

In this experiment, we used oil analysis, a method that is well known and widely used. In this experiment, we found that mayonnaise made from LEY with higher temperatures and longer heating times failed to separate enough oil. Since the sample processing method was applied uniformly, the mayonnaise samples that did not produce sufficient oil after processing were not analyzed further, and their results were not represented. Next, we determined the changes in the acid values of mayonnaise during the storage period (0–8 weeks). The acid values are an indicator of the degree of pre-oxidation of oil. Triglycerides are structurally neutral compounds in which fatty acids and glycerol are linked by ester bonds. However, when oil becomes rancid, fatty acids are released, and free fatty acids can easily oxidize the oil to form free radicals. This has negative effects on oil storage and human health [36]. Based on our data, we found that the acid value of mayonnaise increases with storage time (Figure 5a). At 64 °C, mayonnaise made with thermally processed LEY showed only a slight difference in acid value from the control group. Whereas, at 70 °C, it was observed that all sample groups could not successfully separate enough samples for analysis by week 0–6. Furthermore, it can also be found that mayonnaise made from LEY at high temperature for a long time has better stability and the value is not easily fluctuated. The acid values for all other experimental groups are shown in Appendix A.

The auto-oxidation of unsaturated and polyunsaturated fatty acids in oil is one of the causes of spoilage in mayonnaise. As a result of lipid peroxidation, emulsions that contain fats produce off-flavors and off-odors, thus reducing their shelf lives [37]. Malonaldehyde (thiobarbituric acid, TBA) and peroxide (PV) values are well-established methods to determine oxidation products in fats and oils (either primary or secondary products) [38]. Therefore, we sought to determine the changes of TBA values of mayonnaise during storage. The results show that the TBA value increases with the increase of storage time, which is similar to the trend of acid value (Figure 5b). Additionally, it can be seen that there is only a small difference between experimental and control groups at 64 °C, which agrees with the trend of the acid value. Indeed, the oil and egg yolk can be separated for 5 min and 10 min after six weeks of storage when the temperature is 70 °C. The TBA values for all other experimental groups were shown in Appendix A. It was also found that the mayonnaise has better stability and fever variation. Furthermore, during lipid oxidation, intermediates (free radicals) and products (reactive aldehydes) may interact with other food components (such as proteins, sugars, pigments, and vitamins) to produce negative effects [39]. The increasing stability of emulsions is indicative of slower chemical processes, particularly auto-oxidation [40]. Taken together, our results strongly suggest that mayonnaise made from heated egg yolk has lower acid and TBA values than controls, and that the product is more stable under long term storage.

### 3.10. Sensory Evaluation of Mayonnaise

The quality of mayonnaises prepared from thermally processed LEY is assessed by a panel of 60 panelists using a quantitative descriptive analysis. Prior to each session, the assessors were trained on each attribute, the scale used, and how to score the samples. The average scores of each selected attribute are reported in Table 4. In all five evaluations (appearance, aroma, mouth feel, greasy feeling, overall acceptability), there are no significant differences between the experimental and control groups (*p* > 0.05). Among the experimental groups, the control scored the highest at 5.98, while all experimental groups were lower than the control, but the difference was not statistically significant (*p* > 0.05). With regard to aroma, the experimental groups except at 70 °C/10 min scored higher than its counterpart control group. However, there was no statistically significant difference in scores between the control and experimental groups (*p* > 0.05). In terms of taste, only 65 °C/1 min and 65 °C/10 min were comparable to the control group. In terms of greasy feeling, the 64 °C/1 min and control groups scored slightly higher, but the difference was not significant (*p* > 0.05). In terms of overall acceptance, the mayonnaise experimental groups of 64 °C/1 min and 64 °C/10 min scored higher than the control group. A follow-up conversation with the subjects was also conducted after the evaluation. Most subjects expressed that they were not used to the vinegar taste of mayonnaise. This may be because there is more sugar added to the ingredients of salad dressing in Taiwan, masking the vinegar’s sour taste. Despite this, they all gave 64 °C/10 min a better evaluation than controls. Notably, some subjects reported that the taste of mayonnaise at 70 °C/1 min and 70 °C/10 min was too viscoelastic (compared with commercially available mayonnaise) and they disliked the vinegar taste.

## 4. Conclusions

This study investigated the effects of various physicochemical properties of liquid egg yolk and mayonnaise on the combination of various temperatures (61–70 °C) and thermal processing time (1–10 min), thereby providing an optimal temperature and time for egg yolk sterilization and maintaining mayonnaise quality, which is essential for the egg processing industries. Based on our results, we conclude that the degree of protein denaturation and viscosity were highly correlated with the increases in temperature and time during the thermal processing. In addition, during the thermal processing (61~70 °C for 1–10 min), LEY proteins were neither degraded nor aggregated into macromolecules. Moreover, in high heating conditions, the emulsion stability was higher in both LEY and mayonnaise. The emulsion stability of mayonnaise was significantly (*p* < 0.05) correlated with heating temperature and time, and also significantly better than that of the unheated control group. Furthermore, the rheological analysis revealed that mayonnaise prepared from thermally processed LEY exhibits higher shear stress than that of unheated control group. Results of storage tests have shown that mayonnaise prepared from thermally processed LEY does not influence the quality of the mayonnaise, as evidenced by the fact that the acid values and TBA values were not statistically significant with the unheated control group. Finally, the sensory evaluation results suggest that when compared with the unheated control group, the appearance, aroma, taste, greasy feeling, and overall acceptability of mayonnaise prepared from thermally processed LEY was not statistically significant. Taken together, we here suggest that at the combination of heating conditions above 64 °C/5 min, with adequate sterilization conditions, the LEY can maintain its food processing properties (emulsion stability and fluidity) and be made into mayonnaise products. It does not affect the sensory evaluation of consumers, and it can be produced in the home and used under short-term storage conditions. In terms of storage stability, mayonnaise can be improved by combining heating conditions above 67 °C for 5 min, which is more suitable for factory production. In the future, the combination conditions of different thermal processing of LEY may be applied to various egg yolk products.

## Figures and Tables

**Figure 1 foods-11-01426-f001:**
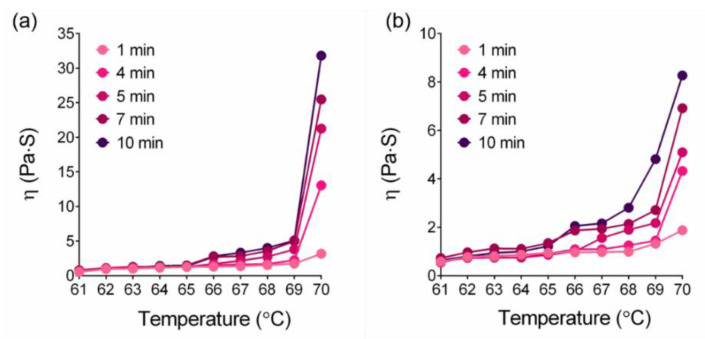
Effect of heating temperature and time on LEY viscosity. (**a**) Shear rate of 10/s. (**b**) Shear rate of 100/s.

**Figure 2 foods-11-01426-f002:**
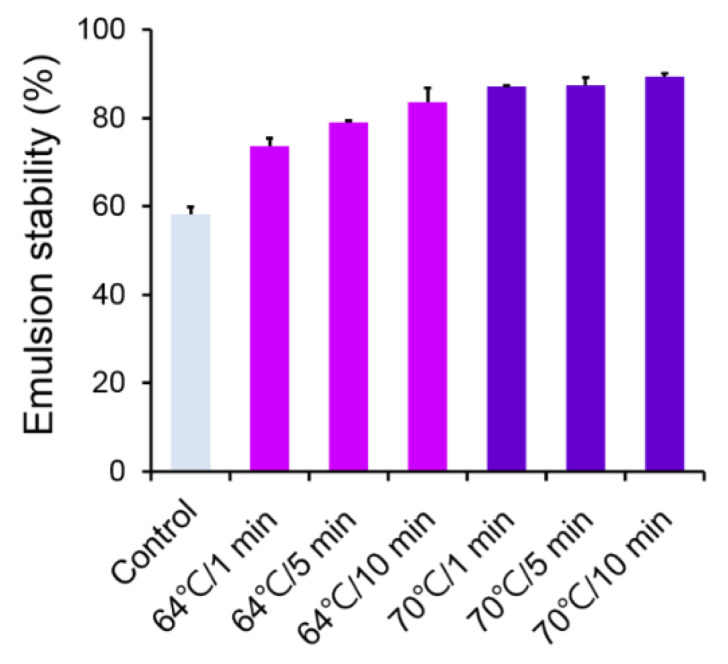
Effect of heating temperature and time on mayonnaise emulsion stability.

**Figure 3 foods-11-01426-f003:**
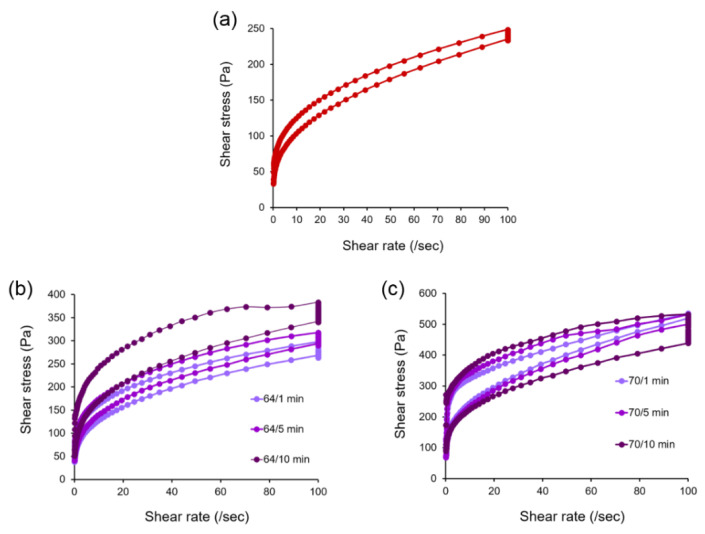
Effect of thermally processed LEY on mayonnaise shear stress. (**a**) Control. (**b**) Temperature at 64 °C for 1, 5, and 10 min. (**c**) Temperature at 70 °C for 1, 5, and 10 min. A scatter chart was used to compare at least two sets of values of data.

**Figure 4 foods-11-01426-f004:**
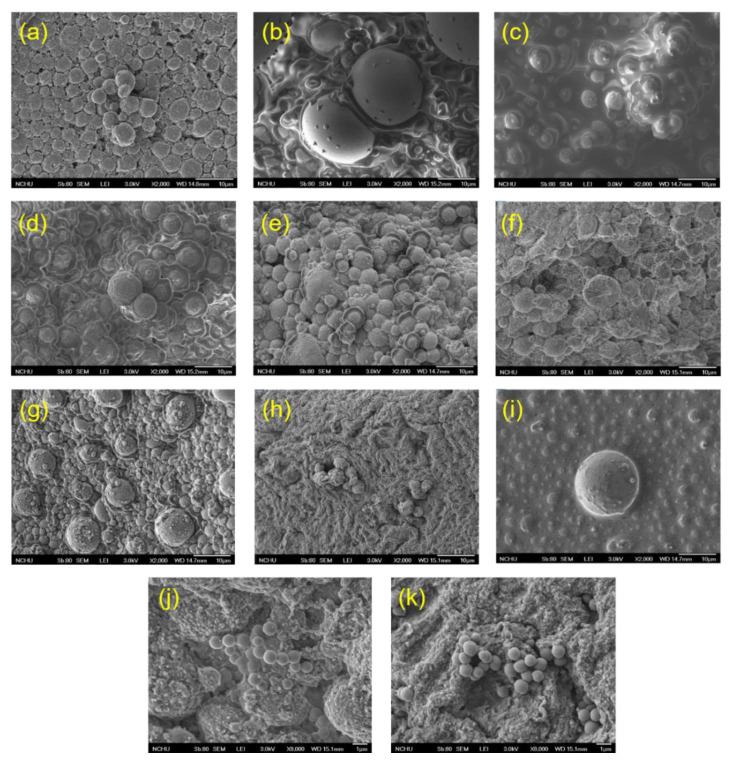
SEM micrographs of mayonnaise. (**a**) Control. (**b**) 61 °C/5 min, (**c**) 61 °C/10 min, (**d**) 64 °C/5 min, (**e**) 64 °C/10 min. (**f**) 67 °C/5 min, (**g**) 67 °C/10 min, (**h**) 70 °C/5 min, (**i**) 70 °C/10 min. Magnification 2000×. (**j**) 67 °C/5 min, (**k**) 70 °C/10 min (at 8000×).

**Figure 5 foods-11-01426-f005:**
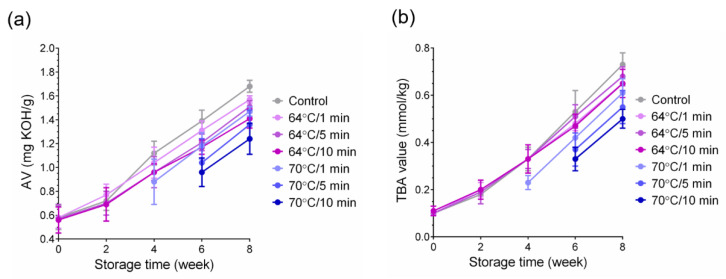
Changes of the acid and peroxidation (TBA) values during storage of mayonnaise products prepared from thermally processed LEY. (**a**) Acid values of temperature at 64 °C and 70 °C for 1, 5, and 10 min. (**b**) TBA values of temperature at 64 °C and 70 °C for 1, 5, and 10 min.

**Table 1 foods-11-01426-t001:** Effect of heat treatment with varying time on LEY protein denaturation.

	Time (min)					
Temperature	0	1	4	5	7	10
Control	1.2 ± 0.7 ^e^	1.2 ± 0.7 ^g^	1.2 ± 0.7 ^c^	1.2 ± 0.7 ^g^	1.2 ± 0.7 ^f^	1.2 ± 0.7 ^f^
61 °C	1.84 ± 0.25 ^de^	2.05 ± 0.37 ^f^	5.15 ± 2.38 ^b^	5.04 ± 1.09 ^f^	7.52 ± 2.21 ^e^	8.22 ± 2.76 ^e^
62 °C	2.14 ± 0.3 ^de^	3.47 ± 0.59 ^cdef^	5.35 ± 0.23 ^b^	5.43 ± 0.2 ^ef^	7.29 ± 1 ^e^	9.75 ± 0.28 ^e^
63 °C	2.07 ± 0.44 ^de^	3 ± 0.02 ^ef^	4.77 ± 0.12 ^b^	7.81 ± 0.5 ^cd^	8.92 ± 0.13 ^e^	10.13 ± 0.71 ^de^
64 °C	1.94 ± 0.68 ^de^	3.39 ± 0.09 ^cdef^	6.11 ± 0.62 ^b^	7.67 ± 0.72 ^de^	8.03 ± 0.46 ^e^	8.66 ± 1.07 ^e^
65 °C	2.42 ± 0.57 ^d^	4.99 ± 0.67 ^cd^	7.11 ± 1.55 ^b^	8.35 ± 0.5 ^cd^	10.15 ± 0.06 ^bcde^	13.61 ± 0.3 ^bc^
66 °C	2.43 ± 0.34 ^d^	4.46 ± 0.83 ^cde^	7.58 ± 0.76 ^b^	8.51 ± 0.5 ^cd^	10.1 ± 0.42 ^bcde^	12.52 ± 0.23 ^cd^
67 °C	2.94 ± 0.59 ^d^	3.52 ± 0.31 ^cdef^	6.82 ± 0.79 ^b^	10.18 ± 0.5 ^bc^	11.15 ± 0.39 ^bc^	13.11 ± 0.6 ^c^
68 °C	4.72 ± 0.45 ^c^	5.22 ± 0.69 ^bc^	6.73 ± 0.04 ^b^	12.15 ± 1.41 ^ab^	10.77 ± 1.07 ^bcd^	13.97 ± 0.59 ^bc^
69 °C	7.12 ± 0.5 ^b^	6.78 ± 0.49 ^ab^	10.85 ± 0.11 ^a^	13.12 ± 2.67 ^a^	12.76 ± 0.47 ^b^	15.8 ± 0.27 ^b^
70 °C	9.11 ± 1.52 ^a^	8.24 ± 1.32 ^a^	11.41 ± 2.17 ^a^	14.13 ± 1.59 ^a^	15.75 ± 1.84 ^a^	18.64 ± 1.33 ^a^

Values (mean ± standard deviation) in the same column followed by different letters differ signifi- cantly at *p* < 0.05 level.

**Table 2 foods-11-01426-t002:** Effect of heat treatment with varying time on LEY protein particle size.

	Time (min)				
Temperature	1	4	5	7	10
Control	6.1 ± 0.9 ^g^	6.1 ± 0.9 ^g^	6.1 ± 0.9 ^d^	6.1 ± 0.9 ^f^	6.1 ± 0.9 ^g^
61 °C	121.4 ± 2.6 ^ab^	115.0 ± 5.2 ^ab^	112.5 ± 15.8 ^bc^	119.1 ± 6.6 ^a^	116.6 ± 7.4 ^bcde^
62 °C	113.5 ± 1.3 ^c^	109.1 ± 6.9 ^abc^	127.3 ± 29.3 ^abc^	125.5 ± 5.2 ^a^	123.1 ± 11.1 ^bc^
63 °C	127.0 ± 4.1 ^a^	34.9 ± 5.4 ^f^	109.9 ± 4.5 ^bc^	52.8 ± 4.8 ^e^	115.7 ± 11.3 ^cde^
64 °C	15.4 ± 1.8 ^f^	108.5 ± 8.2 ^abc^	134.8 ± 15.4 ^ab^	96.0 ± 5.4 ^c^	23.2 ± 3.7 ^f^
65 °C	121.6 ± 2.9 ^ab^	68.4 ± 8.8 ^e^	123.4 ± 7.7 ^abc^	80.9 ± 5.9 ^d^	116.6 ± 7.3 ^bcde^
66 °C	118.9 ± 3.2 ^c^	105.0 ± 5.6 ^bc^	105.5 ± 5.3 ^c^	110.1 ± 3.2 ^b^	121.3 ± 6.1 ^bcd^
67 °C	93.5 ± 7.2 ^f^	85.8 ± 3.7 ^d^	139.8 ± 18.7 ^a^	96.9 ± 3.9 ^c^	104.3 ± 4.1 ^e^
68 °C	104.8 ± 2.0 ^d^	116.0 ± 3.2 ^a^	130.8 ± 10.2 ^abc^	119.7 ± 8.1 ^a^	109.9 ± 4.7 ^ed^
69 °C	120.6 ± 2.0 ^ab^	67.3 ± 5.2 ^e^	118.1 ± 9.9 ^abc^	126.3 ± 2.6 ^a^	129.1 ± 5.0 ^b^
70 °C	101.8 ± 6.9 ^d^	101.7 ± 5.0 ^c^	138.7 ± 6.0 ^a^	102.7 ± 4.1 ^bc^	145.5 ± 6.7 ^a^

Values (mean ± standard deviation) in the same column followed by different letters differ significantly at the *p* < 0.05 level.

**Table 3 foods-11-01426-t003:** Average particle size (μm) of mayonnaise prepared with thermally treated (61~70 °C, 1, 5, and 10 min) LEY.

	Time (min)		
Temperature	1	5	10
Control	4.56 ± 0.82 ^bcd^	4.56 ± 0.82 ^abc^	4.56 ± 0.82 ^bcd^
61 °C	3.68 ± 0.05 ^cd^	3.73 ± 0.42 ^c^	4.65 ± 0.18 ^bcd^
62 °C	5.58 ± 0.64 ^ab^	5.27 ± 1.03 ^ab^	6.84 ± 0.99 ^a^
63 °C	4.43 ± 0.69 ^bcd^	3.46 ± 0.59 ^c^	3.32 ± 0.27 ^bcd^
64 °C	3.55 ± 0.36 ^d^	4.14 ± 0.61 ^bc^	3.38 ± 0.40 ^d^
65 °C	3.70 ± 0.18 ^cd^	3.48 ± 0.07 ^c^	4.96 ± 0.22 ^bc^
66 °C	3.96 ± 0.17 ^cd^	3.83 ± 0.26 ^c^	3.52 ± 0.20 ^bc^
67 °C	3.55 ± 0.58 ^d^	4.04 ± 0.16 ^bc^	3.49 ± 0.25 ^cd^
68 °C	6.50 ± 0.11 ^a^	5.59 ± 0.62 ^a^	4.39 ± 0.42 ^bcd^
69 °C	3.56 ± 0.39 ^d^	3.34 ± 0.32 ^c^	3.79 ± 0.35 ^cd^
70 °C	4.99 ± 1.00 ^bc^	3.32 ± 0.20 ^c^	5.47 ± 1.01 ^ab^

Values (mean ± standard deviation) in the same column followed by different letters differ significantly at *p* < 0.05 level.

**Table 4 foods-11-01426-t004:** Sensory attributes of mayonnaise prepared from thermally processed LEY.

	Sensory Attributes				
Treatment Groups	Appearance	Aroma	Mouth Fell	Greasy Feel	Overall Acceptability
Control	5.98 ± 1.56	5.09 ± 1.56	5.75 ± 1.47	5.62 ± 1.55	5.60 ± 1.57
64 °C/1 min	5.85 ± 1.09	5.22 ± 1.39	5.75 ± 1.45	5.71 ± 1.39	5.82 ± 1.46
64 °C/10 min	5.92 ± 1.14	5.23 ± 1.36	5.75 ± 1.46	5.55 ± 1.38	5.66 ± 1.42
70°C/1 min	5.74 ± 1.15	5.23 ± 1.37	5.72 ± 1.52	5.37 ± 1.45	5.48 ± 1.58
70 °C/10 min	5.71 ± 1.41	4.78 ± 1.44	5.48 ± 1.37	5.49 ± 1.38	5.42 ± 1.43

Values (mean ± standard deviation) in the same column.

## Data Availability

The data presented in this study are available on request from the corresponding author.

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
