# Peer review of "Thermal Processing of Liquid Egg Yolks Modulates Physio-Chemical Properties of Mayonnaise"

_foods, 2022, doi:10.3390/foods11101426_

Round 1

Reviewer 1 Report

The paper reports the results regarding the use of treated egg yolk in mayonnaise phisico-chemical characteristics, evaluating temperature and time as variables. The paper can improve the knowledge on this topic, considering some practical applications, but some steps are requested to improve the scientific report. In particular, some suggestions are proposed, as follows:

-Line 76-80: the discussion of mayonnaise’s shelf life does not consider the presence of antioxidants in the formulation which contrast the lipid oxidation and the successive formation of off-flavours. Authors can also learn more about it by reading https://www.mdpi.com/2304-8158/10/11/2684.

How many replicates were performed for each analysis? Please add this information in the manuscript

Figure 1: It  is difficult to understand the specific differences among variables in the two graphs. So, I propose you to insert a table with post-hoc test to evidence similarity and dissimilarity among results.

Line 349. Delete “<”. If I have well understood, the viscosity did not exceed the value of 5 Pa · S

Line 369. Change the term R2 using the superscript format for 2

Why are reported two curves for each samples in figure 5? It is advisable to describe the mean values.

Line 412. Use the term “among” instead of “between”

How authors explain the higher results on particle size in mayonnaise prepared with LEY treated at 68 °C for 1 min and 62°C for 10 minutes than the other ones?  

Why the sensorial analysis of mayonnaise samples produced LEY treated at 64 °C and 70 °C for 5 minutes was not performed?

Table 3. Delete letters “a” because no statistical differences were noted.

Conclusions. In the paragraph, the comments regarding the higher overall acceptability of samples treated at 64 °C respect to the control is not properly correct. The statistical data elaboration reported in table 3 did not evidence significant differences, so all the samples are similar for all the considered sensorial attributes.

Reviewer 2 Report

Foods

foods-1716496

Thermal processing of liquid egg yolks increased physio-chemical properties of mayonnaise

Dear Editor,

The article deals with the investigation of the effect of various heating temperatures (61-70°C) and times (1-10 min) on physical and chemical properties of liquid egg yolk (LEY) and mayonnaise. The manuscript has been well designed and written. The interpretations are good. My specific comments and questions;

  • Line 49: “salmonella” should be “Salmonella
  • TBA analysis: What about standard curve?
  • Table 1 and 2: Compare the results in terms of not only temperature but also time as statistical.
  • Lines 408-409: Give the possible reason?
  • Figure 8: Give the detail of this figure (score, loading, etc.) At least, please provide biplot figure. What percentage of the difference of data obtained can be explained with this analysis?

Reviewer 3 Report

The manuscript investigates the effects of combination of different temperatures and thermal processing time on different quality parameters of liquid egg yolk and mayonnaise. The obtained results provided possibility for optimization of processing conditions of obtaining best quality for egg processing industries.

Manuscript provides sufficient original material; investigation is extensive and volume of tested quality parameters is sufficient.

Manuscript it is properly written, although English language needs thorough correction.

The title of the manuscript is comprehensive, although needs rephrasing.

Introduction is elaborate enough, with cited corresponding literature.

The Materials and Methods section describes all conducted testing, and it is appropriate.

The results and discussion section presents obtained results and discusses them in detail, although in some sections additional discussion is needed.

Conclusion section is comprehensive enough, and concludes presented results in satisfactory manner

Minor corrections are noted in manuscripts’ pdf file.

Reviewer recommendation: Minor revision.

Round 2

Reviewer 1 Report

The paper has been improved, by the corrections applied by authors according to reviewers’ suggestions.

Only a clarification concerning the reply to Comment 1.

Authors reported that they …” are discussing that the increase of mayonnaise’s self-life is due to its organoleptic properties rather than antioxidant property”.

The proposed food for though was related to the declared shelf life in the text (L76-80) as dependent on “….high viscosity resulting from the close packing of droplets”.

This sentence is not properly correct, because it considers only the physical  parameters. As the authors intend, regarding the overall organoleptic characteristics of mayonnaise, the control of oil oxidation (by additives), microbial counts (by treatments and/or additives) and off-flavour-related production is necessary.

If authors agree can improve finally  this part of introduction.

Author Response

Dear Reviewer-1, many thank you for the suggestion.

Indeed, in the introduction section, we intended to provide the background information related to this study. We are not meant to discuss this information with our results. 

Also, could you help us with how to modify this sentence/phrase?